# Word2vec Skip-Gram Dimensionality Selection via Sequential Normalized Maximum Likelihood

**DOI:** 10.3390/e23080997

**Published:** 2021-07-31

**Authors:** Pham Thuc Hung, Kenji Yamanishi

**Affiliations:** Graduate School of Information Science and Technology, The University of Tokyo, Hongo, Bunkyo-ku, Tokyo 113-8654, Japan; yamanishi@mist.i.u-tokyo.ac.jp

**Keywords:** model selection, information criteria, minimum description length, sequentially normalized maximum likelihood, word embedding, word2vec

## Abstract

In this paper, we propose a novel information criteria-based approach to select the dimensionality of the word2vec Skip-gram (SG). From the perspective of the probability theory, SG is considered as an implicit probability distribution estimation under the assumption that there exists a true contextual distribution among words. Therefore, we apply information criteria with the aim of selecting the best dimensionality so that the corresponding model can be as close as possible to the true distribution. We examine the following information criteria for the dimensionality selection problem: the Akaike’s Information Criterion (AIC), Bayesian Information Criterion (BIC), and Sequential Normalized Maximum Likelihood (SNML) criterion. SNML is the total codelength required for the sequential encoding of a data sequence on the basis of the minimum description length. The proposed approach is applied to both the original SG model and the SG Negative Sampling model to clarify the idea of using information criteria. Additionally, as the original SNML suffers from computational disadvantages, we introduce novel heuristics for its efficient computation. Moreover, we empirically demonstrate that SNML outperforms both BIC and AIC. In comparison with other evaluation methods for word embedding, the dimensionality selected by SNML is significantly closer to the optimal dimensionality obtained by word analogy or word similarity tasks.

## 1. Introduction

In recent years, word2vec has been widely applied to many aspects of Natural Language Processing (NLP) and information retrieval such as machine translation [1,2], text classification [3], text summarization [4], and named entity recognition [5]. Furthermore, word2vec is used in various fields such as materials science [6], healthcare [7], and recommendation engines [8,9,10].

The selection of the dimensionality for word2vec is important with regard to two aspects: model accuracy and computing resources. It is crucial to have a model of dimensionality high enough to learn the regularity of the data, but too high a dimensionality tends to cause overfitting. For instance, the experiment results in [11] demonstrated that the performance of the model in tasks such as Google word analogy, Wordsim353, MTurk771 decreased significantly when the dimensionality increased far from the optimal dimensionality. In addition, a large model is accompanied by a massive number of parameters for storage in the machine during training [12], leading to wasted memory resources. Thus, it is crucial to devise a method that can decide upon a dimensionality that satisfies the ability to capture necessary information from training data as well as makes efficient use of the computational resources.

However, few studies have focused on the dimensionality selection problem. Most research evaluating the effectiveness of word embedding focuses on word analogy and word similarity tasks [13]. These evaluation methods require handcrafted datasets for implementation, but such datasets are currently not available to evaluate model training on non-English verbal and non-verbal data. To the best of our knowledge, only Yin and Shen [11] accomplished the dimensionality selection of a word embedding model without the use of evaluation datasets. However, two aspects of this method need further consideration: the assumption that the noise signal obeys the zero mean-Gaussian distribution has not been verified in real data, and the selected dimensionality is quite different from those obtained by the other evaluation methods based on handcrafted datasets.

Our contributions are two-fold. First, we introduce an effective dimensionality selection method for word2vec based on information criteria. Moreover, our proposed approach does not require handcrafted evaluation datasets, and therefore, can be applied to any type of data not limited to English or verbal-data. This is important and necessary to be able to choose a reasonable dimensionality of the word2vec model when applying it to various fields in information retrieval. Second, from the perspective of information theory, we propose the Sequential Normalized Maximum Likelihood (SNML) criterion in a novel combination with some heuristics for the dimensionality selection problem. The application of our proposed criterion enjoys valuable theoretical guarantees from information theory as well as experimentally ensures that the selected dimensionality is able to capture regularity from the data as well as meet the preferences of models with relatively low but sufficient dimensionality. To the best of our knowledge, this study presents the first application of information criteria in the field of embedding methods as well as the first heuristic comparison of the SNML codelength. These positive results not only encourage wide use of the proposed method in other embedding methods but also suggest a promising solution to evaluate hyper-parameters of a machine learning model.

## 2. Materials and Methods

### 2.1. Related Work

#### 2.1.1. Word Embedding

Representations of words in a vector space have been studied exhaustively in the NLP literature. Beginning with a one-hot vector (the very first representation of words), other word representation methods such as latent semantic analysis [14] and latent Dirichlet allocation [15] have been proposed to improve NLP task performance over time. Various methods that represent words as dense vectors (referred to as “word embedding”), including GloVe [16], word2vec (SG and continuous bag of words) [17], are considered as the state-of-the-art in this field. In this paper, we focus on SG, but the proposed approach can be applied to any other word embedding model.

As the SG model often uses the negative sampling technique, in this study, we work with both original SG (oSG) and Skip-gram with Negative Sampling (SGNS) to clarify the idea behind our approach. In order to apply information criteria on the SG, we summarize it and introduce our notations for both oSG and SGNS.

SG normally takes text data as input to the whole training process. This text data is then processed into pairs of word and context (w,c) in order to feed into a neural network [17]. The preprocess procedure can be applied the same way in the case of other data types. Assume a corpus of words and their contexts are obtained after preprocessing: $D=\left(w,c\right)=\left(w_{1},c_{1}\right),\left(w_{2},c_{2}\right),\ldots ,\left(w_{n},c_{n}\right)$; $w_{i}\in V_{W},c_{i}\in V_{C}$, which are one-hot vectors, where $V_{W}$ and $V_{C}$ are the word and context vocabularies of sizes $S_{W}$ and $S_{C}$, respectively. The training process of oSG attempts to learn the contextual distribution for each word by maximizing the likelihood function seen below.
(1)$$\begin{array}{cc} P_{oSG}\left(c|w;E,F\right) & =\prod_{i=1}^{n}P_{oSG}\left(c_{i}|w_{i};E,F\right)\\ & =\prod_{i=1}^{n}\frac{\exp\left(w_{i}^{T}EF\right)c_{i}}{\sum_{c^{\prime }\in C}\exp\left(w_{i}^{T}EF\right)c^{\prime }}, \end{array}$$
where *E* and *F* are the parameter matrices of the shapes $\left(S_{W}\times d\right)$ and $\left(d\times S_{C}\right)$, respectively. *d* is the dimensionality of the embedding vector space.

Unlike oSG, SGNS learns the probability that a particular context occurred around a word or not: $P(x_{i0}=1|w_{i},c_{i};E,F)$. Furthermore, SGNS introduces negative sampling by sample $S_{z}$ context words: $z_{i}=\left\{z_{i1},z_{i2},\ldots ,z_{iS_{z}}\right\}\in V_{C}^{\left(S_{z}\right)}$ for each particular word $w_{i}$: $P\left(x_{ij}=0|w_{i},z_{ij};E,F\right);j=\left\{1,2,\ldots ,S_{z}\right\}$. The training process of SGNS attempts to maximize the following likelihood function:(2)$$\begin{array}{cc} P_{SGNS}\left(x|w,c,z;E,F\right) & =\prod_{i=1}^{n}P_{SGNS}\left(x_{i}|w_{i},c_{i},z_{i};E,F\right)\\ \; & =\prod_{i=1}^{n}\sigma \left({w_{i}}^{T}EFc_{i}\right)\prod_{j=1}^{S_{Z}}\sigma (-{w_{i}}^{T}EFz_{ij}), \end{array}$$
where σ denotes sigma function [18]. In the remainder of this paper, we denote P(D;θ) for both $P_{oSG}\left(c|w;E,F\right)$ and $P_{SGNS}\left(x|w,c,z;E,F\right)$.

#### 2.1.2. Dimensionality of SG

Unlike our approach, Yin and Shen [11] considered word embedding to be an implicit matrix factorization problem [19] and approached the issue by deciding the rank of the component matrix. Their work was conducted by introducing Pairwise Inner Product (PIP) loss, a measure that evaluates the goodness of the rank of matrix factorization. The best rank is chosen to minimize a given upper bound of the PIP loss.

However, the selected number of dimensions does not agree with the optimal dimensionality performance based on the other evaluation tasks. For example, the best dimensionality of SG chosen by PIP loss is **129**, and the best 5% dimensionalities range from 67 to 218, while the best dimensionalities in the WordSim353 (WS), MTurk771 (MTurk), and Google word analogy (WA) datasets are **56**, **102**, and **220**, respectively [11]. Moreover, the matrix factorization operation conducted during the PIP loss calculation suffers from computational disadvantages and exceeds the calculation limit for huge amounts of data (e.g., Wikipedia dataset in our experiments).

#### 2.1.3. Information Criteria

Word2vec is classified as a self-supervised machine learning model. Therefore, the number of dimensions can be selected by comparing the value of the loss function on the validation dataset. An alternative approach to dimensionality selection involves using information criteria such as the Akaike Information Criterion (AIC) [20], Bayesian Information Criterion (BIC) [21], and Minimum Description Length (MDL) [22]. Compared to the cross-validation method, these information criteria do not require a hold-out validation dataset, which prevents wastage of our precious data.

Since AIC, BIC, and MDL have different backgrounds with regard to the estimation of expected log-likelihood and approximation of the log marginal likelihood, we need to carefully choose the criteria to be used in specific cases. In fact, AIC and BIC rely heavily on the asymptotic theory, which states that as the data size grows to infinity, the estimated parameters converge in probability to the true values of the parameters. However, the asymptotic theory does not apply to word2vec, i.e., as the number of data increases to infinity, we can obtain different optimal parameters set (E, F). Therefore, AIC and BIC are not guaranteed to work theoretically. Nonetheless, several empirical studies have applied them successfully.

Unlike AIC and BIC, MDL with Normalized Maximum Likelihood (NML) codelength is an accurate model selection criterion for real-world data analysis based on limited samples. NML is also known as the best codelength in the context of the minimax optimality property [23].

However, choosing the best method for dimensionality selection is still an experimental task in word2vec. In the next section, we describe in detail the application of MDL to the dimension selection problem and the reason for choosing this method. We then provide empirical comparisons between the methods listed in this section.

In order to apply these information criteria to the dimensionality selection problem, we introduce our notations for the AIC and BIC first.
(3)$$AIC=2\left(S_{W}\times d+d\times S_{C}\right)-2\ln(P(D;\hat{\theta }\left(D\right))),$$
(4)$$BIC=\ln\left(n\right)\left(S_{W}\times d+d\times S_{C}\right)-2\ln(P(D;\hat{\theta }\left(D\right))),$$
where, $\hat{\theta }\left(D\right)=\left(\hat{E}\left(D\right),\hat{F}\left(D\right)\right)$ is the maximum likelihood estimation of the parameters on data D.

### 2.2. Dimensionality Selection via the MDL Principle

#### 2.2.1. Applying the MDL Principle, NML and SNML Codelengths

Word2vec was derived based on the distributional hypothesis of Harris [24], which states that words in similar contexts have similar meanings. Therefore, assuming the existence of the true context distribution for given words $P^{*}\left(\cdot |w\right)$, it is reasonable to choose the dimensionality that has the ability to learn the context distribution most similar to the true distribution. The MDL principle [22] is a powerful solution for model selection, and is considered for the dimensionality selection as per our interest.

The MDL principle states that the best hypothesis (i.e., a model and its parameters) for a given set of data is the one that leads to the best compression of the data, namely the minimum codelength [22]. Specifically, we consider each dimensionality corresponding to a probability model class $M_{d}$.
(5)$$M_{d}=\left\{P\left(D;\theta \right):\;\theta =\left(E\in R^{\left(S_{W}\times d\right)},\;F\in R^{\left(d\times S_{C}\right)}\right)\right\},$$

Assuming that we are able to encode a data series D by a series of only 0 and 1, the length of this binary series is called the codelength of data series D. We take the expression $L\left(D;M_{d}\right)$ as the codelength of data D that can be obtained when encoding with the given information about the model class $M_{d}$. The MDL principle states that the closer the model class $M_{d}$ is to the true distribution generated data $P^{*}\left(\cdot |w\right)$, the shorter the codelength $L\left(D;M_{d}\right)$ that can be obtained.

Given a model class, there are many methods to estimate the shortest codelength of a given dataset such as: two-part codelength, Bayesian codelength [25], NML or SNML codelength. Therein, the NML codelength is the best-known codelength in the MDL literature to achieve the minimax regret [23]. The formula for the NML codelength is given below.
(6)$$L_{NML}\left(D;M_{d}\right)=-\log P(D;\hat{\theta }\left(D\right))+\log C(M_{d}),$$
where $\log C(M_{d})=\log\sum_{D\in D^{\left(n\right)}}P(D;\hat{\theta }\left(D\right))$ is known as Parametric Complexity (PC); $D^{\left(n\right)}$ denotes all possible data series with the length of *n*.

However, the PC term involves extensive computations and is not realistic to implement. Instead, we apply the SNML codelength [26] in this study to reduce the computation cost using the formula seen below:(7)$$L_{SNML}\left(D;M_{d}\right)=\sum_{i=1}^{n}L_{SNML}\left(D_{i}|D^{i-1};M_{d}\right),$$
where $D^{i}$ denotes data series $D_{1},D_{2},\ldots ,D_{i}$ and $D=D_{1},D_{2},\ldots ,D_{n}$. The SNML codelength is calculated as the total codelength where the codelength for each datum is sequentially calculated such as the NML codelength every time it is input. It is known that the SNML codelength is a good approximation to the NML codelength [27]. Since the SNML codelength is sequentially calculated, its computational cost at each step is much lower than that of the NML codelength. Based on the assumption of independence between the data records, the training process of the word2vec model comes after data records have been shuffled. Under this independence assumption, the independent process of SNML does not depend on the order of data.

In addition, the SNML codelength function $L_{SNML}\left(D_{i}|D^{i-1};M_{d}\right)$ can be applied to oGS and SGNS in the forms seen below.
(8)$$\begin{array}{cc} L_{SNML}\left(D_{i}|D^{i-1};M_{d}^{oSG}\right) & =-\log P_{oSG}\left(c_{i}|w^{i},c^{i-1};\hat{\theta }\left(w^{i},c^{i}\right)\right)\\ \; & \;+\log\sum_{c\in V_{C}}P_{oSG}\left(c|w^{i},c^{i-1};\hat{\theta }\left(w^{i},c^{i-1},c\right)\right), \end{array}$$
(9)$$\begin{array}{cc} L_{SNML}\left(D_{i}|D^{i-1};M_{d}^{SGNS}\right) & =-\log P_{SGNS}\left(x_{i}|w^{i},c^{i},z^{i},x^{i-1};\hat{\theta }\left(w^{i},c^{i},z^{i},x^{i}\right)\right)+\\ \; & \;\log\sum_{x\in O^{\left(S_{z}\right)}}P_{SGNS}\left(x|w^{i},c^{i},z^{i},x^{i-1};\hat{\theta }\left(w^{i},c^{i},z^{i},x^{i-1},x\right)\right), \end{array}$$
where $O^{\left(S_{z}\right)}$ is a set of all possible one-hot vectors of $S_{z}$ dimensions.

#### 2.2.2. Some Heuristics Associated with SNML Codelength Calculation

The computation of the SNML codelength still costs $nS_{C}$ times to execute the maximum likelihood estimation for each data record $D_{i}$, which is also not realistic. We introduce two techniques for saving the computational costs for SNML: heuristic comparison and importance sampling on the SNML codelength.
**Heuristic comparison**

A simple observation reveals that if the codelength of data obtained with model class $M_{d}$ is the shortest, then only some part of the data can also be achieved with the shortest codelength compressed with the same model class. Therefore, instead of computing the codelength for all *n* records of data, we can use the codelength of a small set of data. In fact, the results of our experiments show that focusing on the last several thousand records of data is sufficient to compare model classes.

Figure 1 demonstrates the differences in SNML codelengths of different dimensionalities compared with the dimensionality that achieves the shortest codelength on the data. The vertical axis shows the difference of data codelengths obtained by two different dimensionalities shown in the legends (e.g., *d*1 vs. *d*2 *dim*); specifically, it is calculated by $L\left(D^{\prime };d1\right)-L\left(D^{\prime };d2\right)$ where L is the codelength function; $D^{\prime }$ is data; d1 and d2 are dimensionality; while the value of horizontal axis shows the number of records of $D^{\prime }$.

To facilitate comparisons among dimensionalities that are markedly different from one another (such as 30 dimensions versus 65 dimensions in Figure 1(1), or 200 dimensions versus 130 dimensions in Figure 1(2)), it is sufficient to use only 6000 data records to provide information about the best dimensionality to be chosen. Therefore, adding data thereafter simply increases the SNML codelength but does not change our answer substantially. However, for similar dimensionalities, such as 60, 65, and 70 dimensions in Figure 1(3), the first one million data records cannot help us identify the optimal dimensionality. This phenomenon leads to confusion when the codelengths between two model classes are not too different. Furthermore, the number of data records required to determine the best dimensionality comes from the nature of the dataset and the tasks themselves. For example, in the case of word2vec, when SGNS randomizes only a few samples for the negative label from a large context set, the codelength of each data record will vary more than the codelength in oSG, which does not randomly sample negative samples.Therefore, SGNS needs more data records to determine the difference between candidate dimensionality.

To ensure that the correct model is chosen, we need to increase the number of records to estimate the SNML codelength so as to allow a better comparison of these two dimensionalities. However, a small dimension error in the dimensionality selection of word2vec does not affect the final performance considerably. Therefore, the trade-off between the computing time and model selection accuracy is determined by the number of records beyond those required to estimate the SNML codelength with sufficient finality.
**Importance sampling**

Since the size of the context set $S_{C}$ is large (approximately 30,000–100,000 or above, according to the training dataset), the computation of PC for SNML in oSG is still very expensive. We apply the importance sampling method to approximately estimate the SNML description length for each data record. In detail, if a distribution *Q* on the context set satisfies $Q\left(c\right)\neq 0\;\forall \;c\in V_{C}$, the following formula can be applied.

Let $f\left(c\right)=\;P(c|w^{i},\;c^{i-1};\hat{\theta }\left(w^{i},c^{i-1},c\right))$, then
(10)$$\sum_{j=1}^{S_{C}}f\left(c_{j}\right)=E_{Q}\left(\frac{f(c)}{Q(c)}\right)\approx \frac{1}{m}\sum_{c\in S}\frac{f(c)}{Q(c)},$$
where $S=\left\{c_{1},\;c_{2},..,\;c_{m}\right\}\;∼\;Q\left(c\right)$: set of samples draw from distribution Q.

This estimation asymptotes to the true value as *m* (the number of samples) increases, and distribution *Q* is similar to function f(c). In our experiment, the uniform distribution is the best choice for distribution *Q*, and the sampling size is chosen to be 1/10 the size of the context set to balance the computation time and sampling error.

## 3. Results

### 3.1. Experimental Settings

#### 3.1.1. Data

We compared the above-mentioned model selection criteria using SG trained on three datasets: synthetic data, text8, and Wikipedia.
**Synthetic data**

Synthetic data were generated based on several random questions from the WA dataset. Assuming a numeric context set, we generated categorical distributions on this set for all words for which the parameter vectors of the corresponding distributions satisfy the constraints in the questions. For example, corresponding to question: Tokyo,Japan,Paris,France, the process involves the generation of four random contextual distributions, $\tilde{P}\left(\cdot |Tokyo\right)$, $\tilde{P}\left(\cdot |Japan\right)$, $\tilde{P}\left(\cdot |Paris\right)$, $\tilde{P}\left(\cdot |France\right)$, such that:(11)$$cosine\left(\tilde{P}\left(\cdot |Tokyo\right),\tilde{P}\left(\cdot |Japan\right)\right)=cosine\left(\tilde{P}\left(\cdot |Paris\right),\tilde{P}\left(\cdot |France\right)\right),$$

The implementation for generation of such categorical distributions is also available on GitHub (https://github.com/truythu169/snml-skip-gram).

We then sampled words using a uniform distribution and contexts using $\tilde{P}$ adding normal distribution noises. Using these pairs of word and context, oSG and SGNS can be trained to achieve a 100% score on the questions used to create data with the appropriate dimensionality. Furthermore, good dimensionality should result in contextual distributions similar to $\tilde{P}$. To evaluate this similarity, we used a dissimilar function for the oSG model and a similar function for the SGNS model as follows:(12)$$\begin{array}{c} dissimilar\left(M_{d}^{\left(oSG\right)},\tilde{P}\right)=\frac{1}{S_{W}}\sum_{w\in V_{W}}D_{KL}(P_{oSG}\left(\cdot |w;\hat{\theta }\right)||\tilde{P}\left(\cdot |w\right)), \end{array}$$
(13)$$\begin{array}{c} similar\left(M_{d}^{\left(SGNS\right)},\tilde{P}\right)=\frac{1}{S_{W}}\sum_{w\in V_{W}}\rho (f_{P_{SGNS}}\left(\cdot |w;\hat{\theta }\right),f_{\tilde{P}}\left(\cdot |w\right)), \end{array}$$
where, $D_{KL}$ denotes for Kullback–Leibler divergence, ρ denotes Spearman’s rank correlation coefficient, $f_{P_{SGNS}}\left(\cdot |w;\hat{\theta }\right)$ and $f_{\tilde{P}}\left(\cdot |w\right)$ are vectors that take $P_{SGNS}\left(x=1|w,c;\hat{\theta }\right)$ and $\tilde{P}\left(c|w\right)$ ($c\in V_{C}$) as elements, respectively. The choice of $D_{KL}$ for oSG comes from the fact that oSG outputs a categorical distribution, which can be compared with the true distribution using $D_{KL}$; while SGNS results in a list of probability values that are expected to have a strong positive correlation with values of $\tilde{P}$. We used $dissimilar(M_{d}^{\left(oSG\right)},\tilde{P})$ and $similar(M_{d}^{\left(SGNS\right)},\tilde{P})$ as the oracle criterion to evaluate the optimal dimensionality for synthetic data. A good dimensionality selection method is expected to select a dimensionality that is nearby the one chosen by the oracle criterion.
**Text datasets**

The text8 and Wikipedia datasets were preprocessed using a window size of 5, removing words that occur less than 73 times and applying subsampling with a threshold of $10^{-5}$. In addition, we only used the first 20,000 articles of the English Wikipedia dump for the training process.

#### 3.1.2. Training Process


**Optimization settings**


In order to speed up the training process, we implemented a momentum optimizer and mini-batch with a batch size of 1000 for oSG training and stochastic gradient descent for SGNS, as in [28]. A learning rate α for oSG was set to 1.0, and momentum was set to 0.9. For SGNS, α was chosen to be 0.1, and the number of negative samples, 15. The number of epochs was chosen so that the negative log-likelihood value is not significantly reduced. For instance, in the case of oSG, 200 and 90 epochs, respectively, were selected for text8 and Wikipedia, while for SGNS, 15 epochs were selected for text8. Practically, these optimization settings achieve the best performance in our experiment. For example, the best word analogy (WA) scores for text8 are 32.6% (SGNS) and 38.6% (oGS), the corresponding value for Wikipedia is 50.5% (oNS).

Because of the limitations posed by the computational resources, we experimented on a finite number of dimensionalities, which we think is sufficient to clarify the idea behind this research. The evaluated dimensionalities are shown in the figures corresponding to each dataset.
**Importance sampling**

In our experiment, the uniform distribution is the best choice for the distribution *Q* to approximate the SNML codelength. In Table 1, we show the average error of the codelength per record of data according to the sampling size using importance sampling. The implementation is tested on the text8 dataset. Finally, we chose the sampling size to be 1/10 as the size of the context set (the context set comprises about 30,000 words) to balance the computation time and sampling error.
**Estimation of SNML codelength**

The estimation of SNML codelength required us to repeat the parameter estimation $\hat{\theta }\left(w^{i},c^{i}\right)$s×m times, where s is the number of records beyond those required to estimate the SNML codelength, and m is the sampling size. However, repeatedly estimating parameters from scratch is very time consuming. We can alternatively estimate $\hat{\theta }\left(w^{i},c^{i}\right)$ from $\hat{\theta }\left(w^{i-1},c^{i-1}\right)$ by taking the gradient descent of $\left(w_{i},c_{i}\right)$.

### 3.2. Experimental Results

#### 3.2.1. Synthetic Data

We compared five criteria: AIC, BIC, SNML, accuracy on the WA task, and loss value on the validation dataset (CV) with the oracle criterion. The experimental results are shown in Figure 2 and Figure 3. Due to the differences between the criteria values, we scaled all the values to range from 0 to 1 for visual purposes. Moreover, while the dissimilar oracle and other criteria take the dimensionality that minimizes the value, WA takes the maximum. Therefore, in the figure, we draw the line showing the negative value plus one for the dissimilar oracle, AIC, BIC, SNML, and CV so that the higher value states better indicate the dimensionality to be chosen. This scale procedure was also applied to Figure 4, Figure 5, Figure 6, Figure 7, Figure 8 and Figure 9. The horizontal axis in these figures shows the number of dimensions.

The results for oSG show that the BIC exhibits a monotonous decrease, while the optimal dimensionality chosen by the oracle and SNML is 16 and 17, respectively. On the one hand, AIC and CV choose a more distinct dimensionality: 10 and 13, respectively. For SGNS, the oracle chooses 20 dimensions, SNML and CV choose 15, and WA achieves the highest score at 25 dimensions. On the other hand, the BIC chooses 10 dimensions, while the AIC chooses 30.

In both oSG and SGNS, SNML chooses the dimensionality closest to the oracle criterion. Thus, SNML outperforms both AIC and BIC. Note that the synthetic data are designed to achieve a 100% WA score using contextual distribution; however, the scores achieved by using embedded vectors are sensitive to noises and change significantly according to the dimensionality.

#### 3.2.2. Text Data

We compared the SNML criterion with the NLP word analogy task (using WA) and word similarity tasks (using WS, MTurk, and MEN-3k test collection (MEN)). As knowledge regarding the underlying true distribution of the data is lacking, it is difficult to determine the best dimensionality selection method. However, assuming the existence of the true contextual distribution, NLP tasks will roughly prioritize models closest to the true distribution. Therefore, the dimensionality selected by a good method is expected to be close to the optimal dimensionalities for NLP tasks. Note that the evaluation method using scores of NLP tasks is available only in the case of English text data; therefore, it is reasonable to apply a method that achieves the same results as the NLP tasks-based method to any other type of data.

We experimented with at least three runs for each dataset, and the average results are shown in Figure 4, Figure 5 and Figure 6. The comparison of SNML with the information criteria, CV, and PIP is depicted in Figure 7, Figure 8 and Figure 9.

The main results of the study are shown in Table 2, and the actual values of experiments are summarized in the Appendix A. The optimal dimensionalities chosen by the proposed method were compared with the optimal dimensionality in word analogy and word similarity tasks in NLP [13]. Accordingly, for oSG, SNML and CV chose the same dimensionality, which is closer to the optimal dimensionality in NLP tasks than AIC, BIC and PIP. For SGNS, SNML chose the dimensionality closer to the optimal dimensionality for three (WS, WA, MEN) in four tasks (WA, WS, MEN and MTurk) implemented when compared to CV; and four in four tasks implemented when compared to BIC and PIP. We conclude that SNML is better than CV, AIC, BIC and PIP in almost all implemented NLP tasks. Note that we are unable to implement PIP on Wikipedia because the computational complexity was beyond the capabilities of our servers. We are also unable to find the minimum values of BIC and AIC (for text8 train with oSG) for dimensions over a long range.

**Figure 4 entropy-23-00997-f004:**
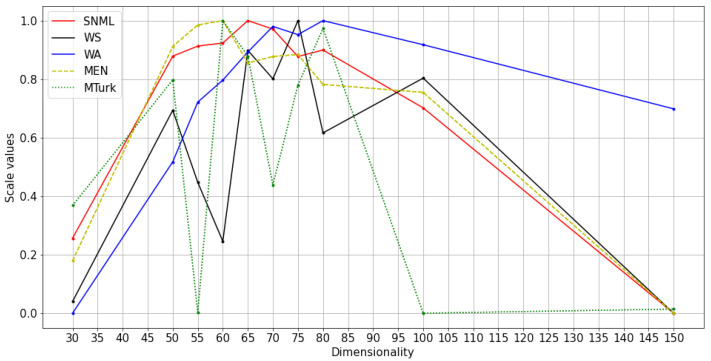
Normalized values and scores on NLP tasks with SNML: text8 training with oSG.

**Figure 5 entropy-23-00997-f005:**
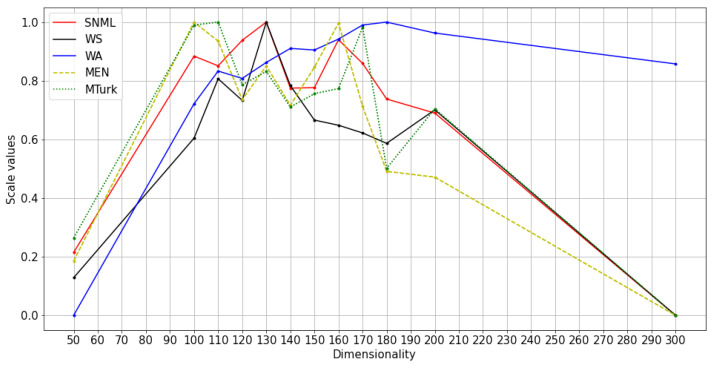
Normalized values and scores on NLP tasks with SNML: Wikipedia training with oSG.

**Figure 6 entropy-23-00997-f006:**
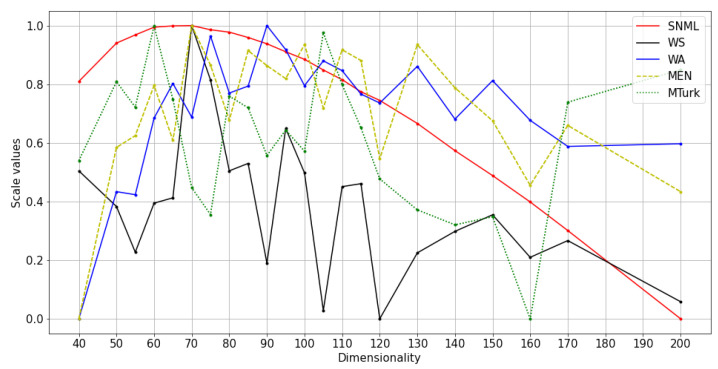
Normalized values and scores on NLP tasks with SNML: text8 training with SGNS.

**Figure 7 entropy-23-00997-f007:**
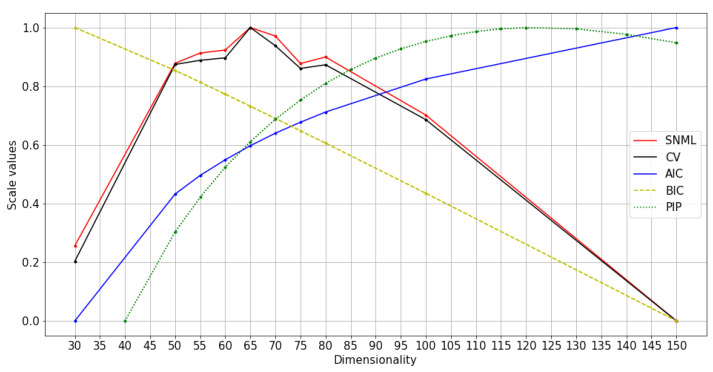
Normalized values of information criteria, CV, and PIP: text8 training with oSG.

**Figure 8 entropy-23-00997-f008:**
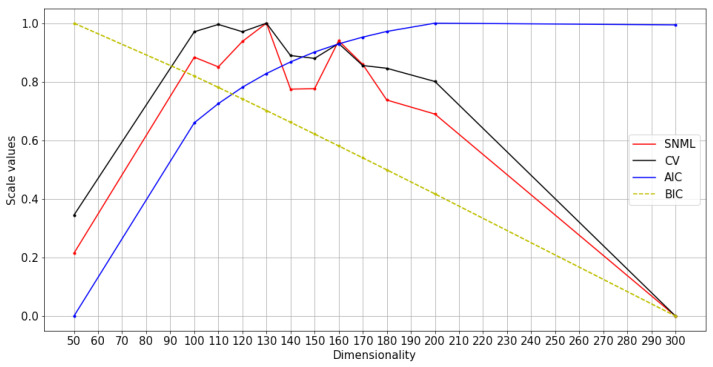
Normalized values of information criteria, CV, and PIP: Wikipedia training with oSG.

**Figure 9 entropy-23-00997-f009:**
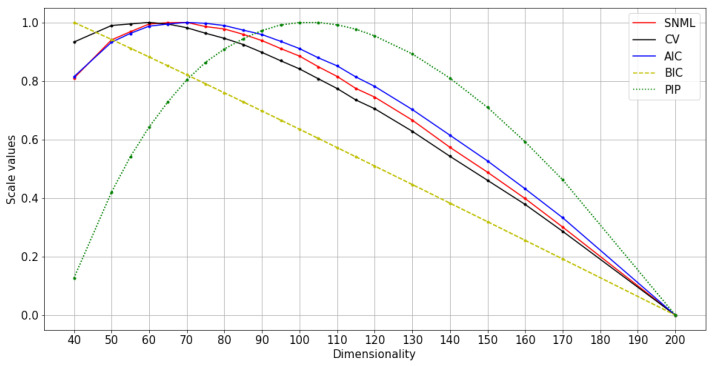
Normalized values of information criteria, CV, and PIP: text8 training with SGNS.

Furthermore, the SNML criterion tends to favor smaller dimensions, although it is sufficient to ensure good performance on other NLP tasks while heavily penalizing model classes that tend to overfitting. This characteristic of SNML helps us avoid choosing large models, and therefore, the available resources should be fully utilized. On the other hand, although AIC favors bigger dimensions and the performance of the model is slightly reduced, this approach is computationally advantageous over SNML. This advantage makes AIC useful in some situations.

## 4. Discussion

To the best of our knowledge, this is the first study that applies information criteria to dimensionality selection for embedding methods.

In order to demonstrate the basic property of our method, we applied it to the very basic model of this field (i.e., Skip-gram model). Our proposed framework can be applied to other embedding methods as well as other neural network-based models once a likelihood function corresponding to any embedding method is defined. For example, in the case of BERT [29], the likelihood function can be defined using a joint probability distribution of a masked token and the next sentence. This likelihood function is then substituted for distribution *P* in Equations (Equation 7)–(Equation 9) to obtain SNML codelengths.

The optimal dimensionality selected by SNML is low; however, it is sufficient to ensure good performance in terms of significant closeness of optimal dimensionality in NLP tasks. However, deep learning models (DL) usually benefit from over-parameterization properties, i.e, the performance of models is not significantly reduced due to the increasing number of parameters. Furthermore, there exist other approaches to the overfitting problem, such as regularization, early-stopping, randomly drop-out, etc., or strategies to adopt large DL models to small data, such as transfer learning, semi-supervised learning, etc. In the paper, we introduce an alternative method to the same problem from an information-theoretic view. The optimal dimensionality selected by the proposed framework can benefit from the over-parameterization property of DL by adopting alternatives such as applying other codelength methods or considering other parameterization methods for the likelihood function. Future challenges in this field include determining which modification results in the most improvement for the dimensionality selection strategy.

## 5. Conclusions

When considering word2vec SG as a probability distribution estimation problem, the optimal dimensionality can provide an estimation of contextual distribution as close as possible to the true distribution-generated data. We tested information criteria (AIC and BIC) and SNML with some heuristics to select such a dimensionality. The experimental results on synthetic data showed that the SNML could choose a dimensionality such that the corresponding probability model is able to learn the contextual distribution closest to the true distribution-generated data. The experiments on text datasets showed that SNML has the ability to choose a desirable dimensionality with regard to two aspects, although low dimensionality is sufficient to ensure good performance in terms of significant closeness of optimal dimensionality in NLP tasks without a hold-out test dataset. Furthermore, SNML typically outperforms AIC, BIC, CV, and PIP in the selection of good dimensionality for NLP tasks in our experiments. Our method therefore holds promise for choosing the most appropriate dimensionality in word2vec when training with data not limited to English or non-verbal.

To the best of our knowledge, this is the first study that applies information criteria to dimensionality selection for word embedding. In fact, the limitations associated with computation or asymptotic estimation of NML or SNML codelength make it difficult to apply such criteria in these areas. By introducing some heuristics in the SNML codelength calculation, we have discovered a new and useful approach, namely MDL-based knowledge embedding. Our proposed approach can be applied to other embedding methods once a likelihood function corresponding to any embedding method is defined. A more detailed evaluation will be left for future study.

## Figures and Tables

**Figure 1 entropy-23-00997-f001:**
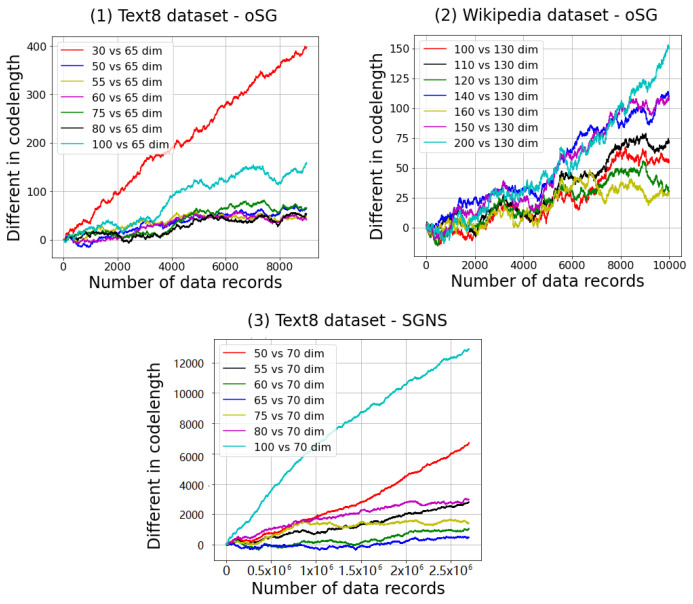
Cumulative SNML codelengths of different dimensionalities compared to the dimensionality result with the shortest codelength.

**Figure 2 entropy-23-00997-f002:**
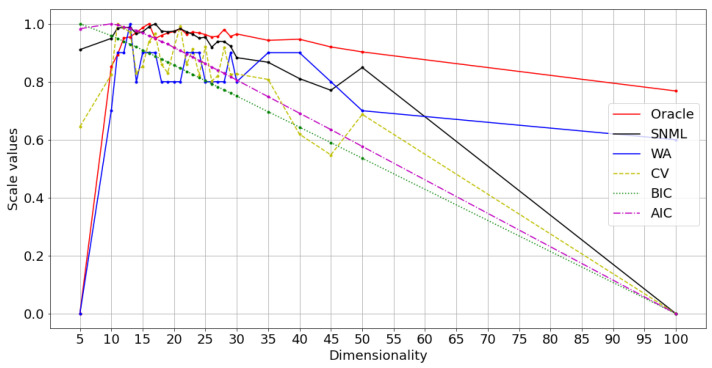
Normalized values of criteria compared with the oracle on artificial data: training with oSG.

**Figure 3 entropy-23-00997-f003:**
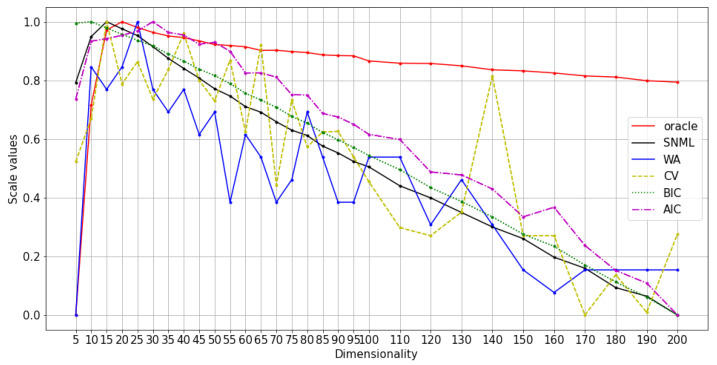
Normalized values of criteria compared with the oracle on artificial data: training with SGNS.

**Table 1 entropy-23-00997-t001:** Average error of importance sampling.

**Sampling Size**	6000	3000	1500	600	300
**Average Error**	0.0022	0.0045	0.009	0.02	0.042

**Table 2 entropy-23-00997-t002:** Optimal dimensionalities chosen by different criteria (-: unknown).

	SNML	WS	WA	MEN	MTurk	CV	AIC	PIP
Text8 (oSG)	65	75	80	60	60	65	-	120
Wikipedia (oSG)	130	130	180	100	110	130	200	-
Text8 (SGNS)	70	70	95	70	60	60	70	105

## Data Availability

(text8) Matt Mahoney, 2011, Large Text Compression Benchmark, http://mattmahoney.net/dc/textdata (accessed on 14 June 2021); (Wikipedia) Wikimedia Foundation, 2019, Wikipedia Database backup dumps, https://dumps.wikimedia.org/ (accessed on 14 June 2021).

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
