# Peer review of "Word2vec Skip-Gram Dimensionality Selection via Sequential Normalized Maximum Likelihood"

_entropy, 2021, doi:10.3390/e23080997_

Round 1
Reviewer 1 Report
General
The authors propose using sequential normalized maximum likelihood as a criterion for selecting the dimension of skip-grams that are commonly used in natural language processing. I find the text relatively well written and clear. My biggest concern is the practical relevance of this work as discussed later.
More detailed comments
I find the language generally good and clear even if some sentence structures strike as bit foreign to me sometimes. (For example the complex sentences without repeating the (admittedly common) subject - “Moreover, our proposed approach does not require handcrafted evaluation datasets therefore can be applied to any type of data not limited to English or verbal-data.” I would probably divide it to two sentences.).
General layout etc. problems
However, the punctuation and layout of equations is very poor. It looks like only the equation 10 (out of 13) is correctly punctuated with the next line correctly (i.e., not) indented. Also the alignment of the equations spanning multiple lines need to be improved. Another trivial source of problems is the layout of figures – they all breach the left margin (which may or may not be acceptable for the initial submission).
Other small problems
-
In equation 2, I do not find using σ for sigmoid-function standard enough for not to be mentioned in the text.
-
In equation (7), doesn’t SNML depend on the order of data? This should be discussed.
-
In 2.2.2, the sentence “… computation … still costs nSC times …” is a bit vague. It would be better to explain that it is the estimation of the maximum likelihood that is in question here.
-
In explaining the results of the Figure 1, the text appears to imply that the reason for the third example requiring 100 times more data is that the difference in candidate dimensions are small. Why wouldn’t the difference rather be in the nature of the dataset or the difference between oSG and SGNS tasks?
-
There is something fishy in the equation 10. If f(c) is a probability of c and we sum over all the possible values of c, shouldn’t we get 1 as a result. Maybe there is logarithm missing there (to get code lengths). Also, I find calling sampling from the uniform distribution “importance sampling” a bit funny.
Significance
The paper is a nice demonstration of the model selection via an information theoretic criterion. However, I wonder if the NLP community is interested in this. With state-of-the-art GPT-3 model featuring 1.75 billion parameters, parsimony appears not to be their main concern.
To authors’ credit, this discrepancy between current deep-learning practices and the model-selection theories is also acknowledged in the discussion. The currently used neural network models are heavily over-parametrized and the regularization is sought by different means such as early-stopping using validation sets, randomly “dropping out” fitted parameters, adding noise to the input, etc. While the task of applying big models for small data (like languages without large text corpora) is well acknowledged, the solution is usually searched from semi-supervised learning, domain adaptation and transfer learning rather than parsimony. And while the dimensions are typical hyper-parameters of the neural network models, selecting them is usually performed by special hyper parameter optimization algorithms that directly try to optimize the performance in the downstream task. For making the models smaller and computationally lighter, special distillation procedures are used to transfer the behaviour of a complex model to a simpler one.
I expect the authors to be well aware of the things mentioned above and more. I also think that dimension selection can possibly be used in many points of the current data processing pipelines. I would encourage authors to elaborate these aspects and possibilities a bit more in their discussion section.
Reviewer 2 Report
The authors presented a method to systematically select the embedding dimension of word2vec skip-gram based on information criterions. Their choice of informatic criterion, Sequential Normalized Maximum Likelihood (SNML) is a computationally fast approximation of the NML criterion, and they show that the number of dimensions selected by SNML is more accurate in synthetic data and in real data compared to the optimal dimensionality obtained by word analogy or word similarity tasks. Overall, the paper is well-written, and explained their method clearly. I do have the following comments:
1. In the synthetic data and text data analysis sections, a more appropriate comparison between methods requires comparison using the same loss. For example, even though AIC selects more dimensions than optimal, that might have a very small effect on the overall performance of the model, compared to the computational advantage it will have over SNML. It will be helpful to have a table with comparisons among computational time/complexities and similar/dissimilar values at the selected dimensions for each method.
2. Need to motivate the problem using evidence. How important is it to select the "optimal" dimensionality? How sensitive is the embedding to the dimensions? Existing literature generally suggests that the performance of the embedding is quite robust to the selection of embedding dimensions. Need some evidence or references that show otherwise.
3. Minor typo: page 8, line 280: Should be (w_i, c_i) instead of (w_i, w_i).
Reviewer 3 Report
Review for the journal “Entropy” (MDPI)
Title: Word2vec Skip-gram Dimensionality Selection via Sequential Normalized Maximum Likelihood
Manuscript number: entropy-1279574
The authors submitted the paper titled “Word2vec Skip-gram Dimensionality Selection via Sequential Normalized Maximum Likelihood” to the journal “Entropy”.
In this work, the authors proposed an information-theoretic method to evaluate the dimensionality for word2vec, a well-known machine learning tool used for natural language analysis (classification, summarization, etc.). The proper evaluation of the dimensionality for word2vec is essential since it defines both the performances and the computational load required. A too-small dimensionality would cause small performances, while too large will involve overfitting-related issues and too high computational time (even so large to make the word2vec application unsuitable). Normally, the dimensionality selection can be performed by previous experience or ad hoc analyses on synthetic or known cases. The approach used by the authors is based on the Sequential Normalized Maximum Likelihood (SNML).
The paper is well organised and uses the classical structure: Introduction, Methods, Results, Discussion and Conclusions. The introduction is quite brief, but it offers a general overview of the literature and a clear statement of the problem. The methodology is also well written and organised, I was able to follow the section without particular efforts. The only change that I would suggest is to put subsection 3.1 (experimental settings) of the result section into the method section (since you are still describing the materials and methods, and not the results). Also, the other sections are well presented and I have not other comments regarding the structure of the paper.
The results are performed with three datasets, a synthetic one, and two experimental (Text8 and Wikipedia). Regarding this section I have some comments/suggestions which I would implement before the publications:
- Figure 2, right: the maximum values of all methods are around 10 – 15, but I see that your dimensionality resolution is 5 (right?). I suggest to increase the resolution to one in this region (let us say from 5 to 25) because It would be useful to understand the degree of difference. For example, you say that the BIC value is 10, while oracle and SNML are 15. But since the resolution is 15, it may be that oracle is 13, BIC is 12 and SNML is 17, involving opposite results to the one declared in the paper.
- Figure 3: same of figure 2, I suggest to increase the resolution in the critical region to better understand the differences. Again, for Text8 (oSG), SNML returns 65 and MEN 60, but it may be that SNML true value is 69 and MEN is 61, or that SNML and MEN are both 63.
- As you say, dimensionality is important to characterise the performances and computational load of the method. However, I was not able to find in the results how these two characteristics change according to the dimensionality. For example, for Text8 (oSG), SNML return 65 and MEN 60. How do the performances change in these two cases? And the computational load? Just a rough estimation for one of the cases may be enough.
Other minor comments:
- Table 2: You may remove the BIC column since it is not evaluated.
Round 2
Reviewer 3 Report
Dear Authors and Editor,
thank you for answering my questions. I feel the paper is now suitable for publication.